# Core–Shell CoS_2_@MoS_2_ with Hollow Heterostructure as an Efficient Electrocatalyst for Boosting Oxygen Evolution Reaction

**DOI:** 10.3390/molecules29081695

**Published:** 2024-04-09

**Authors:** Donglei Guo, Jiaqi Xu, Guilong Liu, Xu Yu

**Affiliations:** 1Key Laboratory of Function-Oriented Porous Materials, College of Chemistry and Chemical Engineering, Luoyang Normal University, Luoyang 471934, China; gdl0594@163.com (D.G.); 18836032666@163.com (J.X.); glliu@tju.edu.cn (G.L.); 2Institute of Innovation Materials and Energy, School of Chemistry and Chemical Engineering, Yangzhou University, Yangzhou 225002, China

**Keywords:** molybdenum disulfide, core–shell structure, heterostructure, oxygen evolution, Prussian blue

## Abstract

It is imperative to develop an efficient catalyst to reduce the energy barrier of electrochemical water decomposition. In this study, a well-designed electrocatalyst featuring a core–shell structure was synthesized with cobalt sulfides as the core and molybdenum disulfide nanosheets as the shell. The core–shell structure can prevent the agglomeration of MoS_2_, expose more active sites, and facilitate electrolyte ion diffusion. A CoS_2_/MoS_2_ heterostructure is formed between CoS_2_ and MoS_2_ through the chemical interaction, and the surface chemistry is adjusted. Due to the morphological merits and the formation of the CoS_2_/MoS_2_ heterostructure, CoS_2_@MoS_2_ exhibits excellent electrocatalytic performance during the oxygen evolution reaction (OER) process in an alkaline electrolyte. To reach the current density of 10 mA cm^−2^, only 254 mV of overpotential is required for CoS_2_@MoS_2_, which is smaller than that of pristine CoS_2_ and MoS_2_. Meanwhile, the small Tafel slope (86.9 mV dec^−1^) and low charge transfer resistance (47 Ω) imply the fast dynamic mechanism of CoS_2_@MoS_2_. As further confirmed by cyclic voltammetry curves for 1000 cycles and the CA test for 10 h, CoS_2_@MoS_2_ shows exceptional catalytic stability. This work gives a guideline for constructing the core–shell heterostructure as an efficient catalyst for oxygen evolution reaction.

## 1. Introduction

Exploring clean and renewable energy to substitute traditional fossil fuels is important to alleviate the current global environmental problems [1,2]. Hydrogen energy with high gravimetric energy density and pollution-free characteristics has been considered as a potential energy source in many applications [3,4]. Oxygen evolution reaction (OER), as a half-reaction of electrocatalytic water splitting, is a complicated four-electron transfer process, and the sluggish kinetics still need to be improved to enhance the efficiency of water electrolysis [5]. Conventional water electrolysis often relies on noble metals, such as platinum and ruthenium-based catalysts [6,7,8,9], which exhibit fast kinetic behavior and low overpotential. Despite these merits of noble-metal catalysts, the high prices and rarity of resources severely inhibit their widespread application [10], and the development of cheap and effective catalysts has attracted attention to improve the dynamic reaction [11].

Recently, transition-metal-based composites with earth-abundant sources and high efficiency have been proposed to substitute noble-metal catalysts. Many kinds of metal composites, including metal sulfides [12,13,14,15], metal phosphides [16,17], and metal oxides/hydroxides [18,19,20], have been demonstrated to show improved catalytic OER activity in an alkaline electrolyte. As confirmed by computational and experimental analyses, cobalt sulfides with a refined surface chemical state can be used as effective catalysts to facilitate the electrochemical reaction during the OER process [21,22]. Apart from modifying the composition, a delicate structural design is also important for improving the electrocatalytic performance. Great attention has been paid to develop Prussian blue analog (PBA)-based nanomaterials in the application of energy storage and conversion systems [23,24], while the poor electrical conductivity and inertness of intrinsic activity prevent the application of pristine PBA nanomaterials for OER [25]. The regulation of the adsorption energy of active sites and the improvement of intrinsic conductivity can be modified by the incorporation of heteroatoms (S, P, Se, F, etc.) [26,27,28,29]. Metal sulfides derived from the Prussian blue analog can hold structural and compositional merits, and a hollow structure can be derived by ion exchange during the hydrothermal reaction process, which is favorable for accelerating electrolyte ion diffusion and facilitating charge transfer [25,27,30]. PBA derivatives with porous structures can expose the large specific surface area and increase the utilization of active sites. Furthermore, introducing the second functional into the PBA derivatives is a promising strategy to improve the catalytic OER activity of catalysts.

Transition-metal dichalcogenides (TMDCs) with layered structures have been proposed as economical and efficient catalysts, and have been proven to have highly efficient activity in the dominant electrochemical reaction process of oxygen evolution reaction [31,32]. Most possible reasons for this include the exposure of active basal and edge sites to accelerate the reaction kinetics and the modification of the intrinsic activity of layered nanosheets. Many efforts have been made to reduce the charge transfer resistance of TMDCs by the incorporation of heteroatoms and the formation of heterostructures [21,33,34]. Molybdenum disulfide (MoS_2_), with a metallic 1T crystal structure, shows improved catalytic OER performance by activating intrinsic activity on the basal plane and edge sites [35,36,37]. The surface’s defective sites can be modified to change the electronic structure and boost the active sites. However, the restacking and aggregation of layered MoS_2_ are still key issues to be solved. The core–shell morphology of the catalyst can show specific physical properties and surface chemistry, which is beneficial to expose the large surface area and active sites, effectively protect the active surface, inhibit the aggregation of layered material, and optimize the electronic structure, resulting in the improvement of catalytic behaviors [38]. Therefore, the construction of a heterostructure can further improve the catalytic OER performance by forming strong interfaces between two different transition metal components [39,40,41], while the controllable morphology of the heterostructure can efficiently prevent the aggregation of layered TMDCs [42,43]. Despite the above issues, we aimed to construct a layered TMDC-based catalyst to develop the OER performance of a compositional-adjusted CoS_2_@MoS_2_ heterostructure.

In this work, we prepared a hollow CoS_2_@MoS_2_ heterostructure with a core–shell structure through hydrothermal and sulfidation processes. The vertically aligned structure, revealed by microscopic analysis, guarantees the distribution of MoS_2_ nanosheets without aggregation and provides abundant channels for electrolyte diffusion. The surface chemistry of CoS_2_@MoS_2_ is tuned by the existence of 1T-phase MoS_2_ and a strong electrochemical interaction at the CoS_2_/MoS_2_ interface. On account of the heterostructure and modified chemical state, CoS_2_@MoS_2_ shows excellent catalytic OER performance, such as low overpotential, a small Tafel slope, low charge transfer resistance, and good electrochemical stability.

## 2. Results and Discussion

### 2.1. Synthesis and Characterization of CoS_2_@MoS_2_

As illustrated in Figure 1a, the cobalt-sulfide-coupled molybdenum disulfide (CoS_2_@MoS_2_) nanocube with a core–shell structure was prepared using co-precipitation, low-temperature sulfurization, and hydrothermal methods. In brief, the cobalt Prussian blue analog (Co-PBA) was initially synthesized using the precipitation method at room temperature and further treated by low-temperature sulfurization to obtain the CoS_2_ nanocube. After mixing with the Mo precursor, the interlaced MoS_2_ nanosheets were grown on the surface of the CoS_2_ nanocube during the hydrothermal process, and the core–shell CoS_2_@MoS_2_ catalyst was finally obtained. Scanning electron microscopy (SEM) was carried out to characterize the morphology of the CoS_2_@MoS_2_. As shown in Figure 1b, the surface of the Co-PBA nanocube is smooth and the average size of the diameter is about 200 nm. After the initial sulfurization, the CoS_2_ showed a hollow structure arising from the ion exchange of Co, and the roughness of the surface increased due to the formation of metal sulfide nanoparticles (Figure 1c). Furthermore, MoS_2_ nanosheets were vertically aligned on the CoS_2_ nanocube, which efficiently inhibited the aggregation of MoS_2_ nanosheets, as shown in Figure 1d. It can be found that the average diameter of CoS_2_@MoS_2_ is larger in contrast to the hollow CoS_2_ nanocube due to the growth of MoS_2_ on the surface. The increased roughness and porous surface are favorable as they provide a large surface area and sufficient channels for electrolyte ion diffusion.

The morphology of CoS_2_@MoS_2_ was further investigated using transmission electron microscopy (TEM), as shown in Figure 2. Figure 2a shows the TEM image of pristine Co-PBA with a transparent morphology and smooth surface, which is consistent with the SEM result. After sulfurization treatment, the hollow structure of the CoS_2_ nanocube can be observed, and the rough surface is composed of metal sulfides (Figure 2b). As shown in Figure 2c, the MoS_2_ nanosheets are uniformly grown and distributed on the CoS_2_ surface to form the core–shell structure. From the high-magnification TEM image in Figure 2d, two different lattice fringes can be observed, which are intertwined with each other, and the values of interplanar spacing of 0.63 and 0.20 nm correspond to the (002) plane of MoS_2_ and the (220) plane of CoS_2_. In particular, the interplanar spacing of MoS_2_ for CoS_2_@MoS_2_ is larger than that of pristine MoS_2_, and the expanded d-spacing results from the hybridization of CoS_2_ with MoS_2_ nanosheets.

X-ray diffraction was applied to evaluate the crystal structure of the CoS_2_@MoS_2_, as shown in Figure 3a. The apparent characterization peak at ~14.1° corresponds to the (002) plane of MoS_2_ [33], and another three peaks are indexed to the presence of CoS_2_ [PDF#41-1471], implying that CoS_2_@MoS_2_ consists of MoS_2_ and CoS_2_. X-ray photoelectron spectroscopy (XPS) was used to evaluate the surface chemical state and electronic structure of CoS_2_@MoS_2_, and the C 1s peak at 284.6 eV was used to calibrate all of the spectra. CoS_2_@MoS_2_ consists of Mo, C, O, Co, and S elements (Figure 3b), and the existence of oxygen is possible due to the surface oxidation that occurs from the adsorption of water when the sample is exposed to an air atmosphere. As the high-resolution Co spectra of CoS_2_ show in Figure 3c, the one pair of fitted peaks at 781.7 and 785.1 eV are indexed to the Co^3+^ and Co^2+^ of the spin-orbital of Co 2P_3/2_; another pair of peaks at 798.0 and 800.6 eV correspond to the Co^3+^ and Co^2+^ of the spin-orbital of Co 2p_1/2_, and the additional peaks at 788.4 and 803.5 eV are the related satellite peaks, respectively. In comparison to CoS_2_, two pairs of peaks for CoS_2_@MoS_2_ at 785.7/801.2 eV and 782.2/798.6 eV correspond to the Co^2+^ and Co^3+^ of the spin-orbital of Co 2p_3/2_ and Co 2p_1/2_, with related satellite peaks at 788.9 and 804.5 eV. It can be seen that the peaks for CoS_2_@MoS_2_ positively shifted to the high-energy region, while the Co 2p satellite peaks of CoS_2_/MoS_2_ are stronger than those of the pristine CoS_2_, implying that the high-spin Co^3+^ is dominant [44]. The Mo 3d spectra can be deconvoluted to five peaks, as shown in Figure 3d. A pair of peaks at 229.2 and 232.3 eV is indexed to the 1 T of Mo 3d_5/2_ and Mo 3d_3/2_, two peaks at 229.9 and 233.2 eV are ascribed to the 2H phase of Mo 3d_5/2_ and Mo 3d_3/2_, and the additional peak at 236.1 eV corresponds to the oxidized Mo species, respectively [45]. The base of MoS_2_ with the 2H phase is a sulfur-modified surface, which has poor catalytic activity [46], and the metallic 1T phase of MoS_2_ can expose more active sites and improve the electronic conductivity [47]. The peak area of the 1T phase is larger than that of the 2H phase, implying that the metallic 1T phase is dominant for the pristine MoS_2_. In comparison, CoS_2_@MoS_2_ shows a negative shift of Mo 3d peaks to the low-energy region. The negative shift of Mo 3d and the positive shift of Co 2p can be attributed to the strong chemical interaction at the CoS_2_/MoS_2_ interface by forming a Mo-S-Co bond [48]. Figure 3e shows the S 2p spectra of all catalysts and CoS_2_@MoS_2_ shows two peaks at 161.7 and 162.8 eV, corresponding to the spin-orbital of S 2p_3/2_ and S 2p_1/2_. The peaks of CoS_2_@MoS_2_ are positively shifted in contrast to CoS_2_, and the peak at high binding energy is decreased, implying the strong interaction at the CoS_2_/MoS_2_ interface. The deconvoluted O 1s spectra can be deconvoluted into three dominant peaks, as shown in Figure 3f, including the metal–oxide bond (lattice oxygen, O^2−^), defective oxygen with low oxygen coordination (oxygen vacancies, O^−^), and the hydroxyl species or adsorbed oxygen on the surface (OH^−^/O_2_) [49]. 

### 2.2. Electrocatalytic Performance

The electrocatalytic OER performance of CoS_2_@MoS_2_ was evaluated via a three-electrode configuration in 1 M KOH electrolyte, and the electrolyte was treated by flowing nitrogen gas before measurement. Two additional catalysts of MoS_2_ and CoS_2_ as control samples were analyzed under the same conditions and all the overpotentials were calibrated after IR correction. Figure 4a shows the polarization curves of all catalysts at 5 mV s^−1^. To reach the current density of 10 mA cm^−2^, CoS_2_ and MoS_2_ show moderated overpotentials of 304 and 381 mV, and CoS_2_@MoS_2_ shows the highest OER performance with the smallest overpotential of 254 mV. At the high current density of 20 and 80 mA cm^−2^, CoS_2_@MoS_2_ exhibits overpotentials of 278 and 346 mV, which are both smaller than those of the control samples (Figure 4b). The Tafel slope, as an important factor in evaluating the kinetic HER behavior, was calculated by fitting the selected region of polarization curves in Figure 4c. The corresponding Tafel slope of CoS_2_@MoS_2_ is 86.9 mV dec^−1^, which is much lower than that of pristine CoS_2_ and MoS_2_, implying its fast dynamic behavior and good catalytic performance. The value of the Tafel slope is between 40 and 120 mV dec^−1^, revealing that the Volmer–Heyrovsky process is dominated as the rate-determined step [50]. Furthermore, electrochemical impedance spectroscopy (EIS) was carried out to further confirm the kinetic behavior of CoS_2_@MoS_2_ and unveil the high OER activity. Nyquist plots were deconvoluted by the equivalent circuit (inset of Figure 4d), in which Rs, R1, and Rct represent the solution resistance, the contact resistance on the surface of the electrode, and the charge transfer resistance. CoS_2_@MoS_2_ shows the smallest diameter of a semi-circle in the high-frequency region and the lowest Rct value (Figure 4d), indicating that CoS_2_@MoS_2_ has good electrical conductivity, permitting it to improve the electron transfer capability. The features of the Tafel slope and EIS analysis strongly prove that CoS_2_@MoS_2_ exhibits fast catalytic kinetics and exceptional catalytic OER performance owing to the core–shell structure and strong chemical coupling at the CoS_2_@MoS_2_ interface. In comparison, the catalytic OER activity of the core–shell CoS_2_@MoS_2_ catalyst with a low overpotential is also superior to the reported results shown in Figure 4e [22,51,52,53,54,55,56,57,58].

The kinetic behavior of the CoS_2_@MoS_2_ catalyst was further confirmed by active energy derived from the Arrhenius equation, and the related CV curves were measured at different temperatures of 25 °C, 35 °C, 45 °C, 55 °C, and 65 °C. As shown in Figure 5a, the overpotential of the CoS_2_@MoS_2_ catalyst gradually decreases along with the increased temperature, implying that the OER activity is significantly improved. Meanwhile, the CoS_2_@MoS_2_ catalyst at 65 °C shows the lowest Tafel slope value, indicating fast dynamic properties with the increase in temperature (Figure 5b). The active energy can be determined by fitting the 1/T vs. log (j) [59], and the value of log (j) is obtained from the intersection where the Tafel slope intersects with the x-axis. The value of active energy is 55.1 kJ mol^−1^ according to the linear fitting of the Arrhenius plot, as shown in Figure 5c.

To evaluate the catalytic behavior of the CoS_2_@MoS_2_ catalyst, the CV curves in the non-faradic reaction area were measured at a scan rate from 5 to 50 mV s^−1^ (Figure 6a–c), and the related double-layer capacitance (Cdl) was calculated by fitting the CV curves shown in Figure 6d. The value of Cdl for CoS_2_@MoS_2_ is 10.9 mF cm^−2^, which is much higher than that of MoS_2_ (0.65 mF cm^−2^) and CoS_2_ (5.8 mF cm^−2^), predicting that the CoS_2_@MoS_2_ catalyst can expose abundant active sites on the surface to improve the electrocatalytic OER performance. The vertical growth of MoS_2_ nanosheets on the surface of the CoS_2_ nanocube not only prevents the restacking of layered MoS_2_, but also exposes more marginal active sites. The formed core–shell structure can provide many pathways for electrolyte diffusion and charge transfer and improve the utilization of active sites. The catalytic stability is an important consideration to evaluate the electrochemical performance of the catalyst during the OER process. The polarization curves of CoS_2_@MoS_2_ are initially measured by CV at 5 mV s^−1^ (Figure 6e), and almost no change of the curves at the 1st and 1001st cycles can be observed, revealing the high catalytic OER stability favored by the core–shell structure and CoS_2_@MoS_2_ formation. Meanwhile, a chronoamperometry (CA) test was further applied at the overpotential of 254 mV for 15 h (Figure 6f), and there was no apparent change in current density during the continuous operation, further demonstrating the exceptional catalytic stability.

The excellent electrocatalytic OER activity of CoS_2_@MoS_2_ in alkaline electrolytes can be ascribed to the synergistic effect of the core–shell structure and the generation of a heterostructure interface; the possible reasons for this are as follows. (1) The hollow CoS_2_ with a nanocubic structure can act as the carrier to prevent the restacking of MoS_2_ nanosheets and provide the pathways for fast electrolyte ion diffusion. (2) Layered MoS_2_ nanosheets are vertically grown on the surface of CoS_2_, the utilization of exposed marginal and basal active sites of CoS_2_@MoS_2_ can be improved, and the contact between the electrolyte and the electrode can be increased. (3) The core–shell structure, with CoS_2_ as the core and MoS_2_ as the shell, can provide abundant exposed active catalyst sites, improve the reaction rate, inhibit the aggregation of catalyst, accelerate water oxidation, and promote the adsorption/desorption of intermediates. (4) The different chemical components and crystal structure can induce the lattice strain to affect the absorbability of intermediates, and the charge transfer at the interface can adjust the electronic structure of the catalyst. (5) The strong chemical interaction at the CoS_2_@MoS_2_ interface can boost the active sites to improve the catalytic OER activity. 

The change in the morphological structure and surface chemistry of CoS_2_@MoS_2_ after CA measurement in 1 M KOH was unveiled by SEM, TEM, and XPS. After the catalytic OER test, the roughness of the CoS_2_@MoS_2_ surface was observed and the transparency of the catalyst decreased, which was attributed to the formation of metal oxide/hydroxides and small residual components on the surface (Figure 7a,b). Figure 7c shows the high-resolution Co 2p spectra of CoS_2_@MoS_2_ before and after the CA test, and the major peaks of Co 2p_3/2_ and Co 2p_1/2_ shifted to the low-energy region arising from the surface oxidation of CoS_2_@MoS_2_ during the CA test using alkaline electrolytes, such as the intermediates of cobalt oxides or oxyhydroxides. In particular, the peaks of CoS_2_@MoS_2_ become broad and negatively shift after the CA test in contrast to its pristine state, which is attributed to the change in the surface chemistry through the conversion of the metal-S bond to the metal-O bond during the stability test. Meanwhile, the Mo 3d spectra show two primary peaks of Mo 3d_5/2_ at 229.2 eV and Mo 3d_3/2_ at 232.1 eV, and an additional peak at 236.0 eV attributed to the formation of the Mo-O bond after the CA test (Figure 7d). The slight shift of the Mo 3d peaks and the presence of a strong Mo-O bond imply that the surface of the CoS_2_@MoS_2_ is oxidized.

## 3. Materials and Methods

### 3.1. Synthesis of Co-PBA

First, 150 mg of cobaltous chloride hexahydrate was mixed with 900 mg of sodium citrate dehydrate in 50 mL of deionized (DI) water under magnetic stirring for 30 min. Then, 50 mL of potassium hexacyanocobaltate (III) solution (about ~300 mg) was added into the above mixture drop by drop within 10 min under continuous stirring. After continuous aging at room temperature for 24 h, the solution was treated by centrifugation and the obtained precipitates were washed several times with DI water/ethanol. The precipitate was dried at 60 °C overnight and denoted as Co-PBA.

### 3.2. Synthesis of CoS_2_

In total, 50 mg of the as-obtained Co-PBA was dispersed in 15 mL of ethanol under magnetic stirring for 1 h to obtain the dispersion. Then, 15 mL of sodium sulfide solution (4 mg mL^−1^) in DI water was subsequently added into the dispersion under magnetic stirring for 10 min. The mixture was poured into a 50 mL Teflon-lined stainless-steel autoclave and heated at 120 °C for 10 h. After cooling down to room temperature, the precipitate was centrifuged and washed several times with water/ethanol. Finally, the precipitate was thermally activated at 400 °C for 1 h under a N_2_ atmosphere.

### 3.3. Synthesis of Pure MoS_2_

In this step, 150 mg of thiourea and 80 mg of ammonium molybdate were dissolved in 30 mL of DI water under magnetic stirring for 1 h. The mixture was subsequently transferred into a 50 mL Teflon-lined stainless-steel autoclave and heated at 240 °C for 20 h and naturally cooled down to room temperature. The precipitate was centrifuged and washed several times with water/ethanol. Finally, the precipitate was thermally annealed at 400 °C for 1 h under a N_2_ atmosphere.

### 3.4. Synthesis of Hollow CoS_2_@MoS_2_

Firstly, 50 mg of Co PBA was dispersed in 12 mL of ethanol under ultrasonication and 15 mL of sodium sulfide solution (4 mg mL^−1^) in DI water was subsequently added into the dispersion under magnetic stirring for 10 min. Then, 150 mg of thiourea and 80 mg of ammonium molybdate were dissolved in 30 mL of DI water under magnetic stirring for 1 h, and the mixture was transferred into a 50 mL Teflon-lined stainless-steel autoclave and heated at 120 °C for 5 h and 240 °C for 20 h. The black precipitate was centrifuged and washed with DI water/ethanol following air-drying at 60 °C overnight. Finally, the product was obtained by thermal annealing at 400 °C for 1 h under a N_2_ atmosphere and denoted as CoS_2_@MoS_2_.

### 3.5. Characterization

Powder X-ray diffraction (XRD) patterns were recorded on a Bruker D8 Advance powder X-ray diffractometer using a Cu Kα (λ = 1.5405 Å) radiation source operating at 40 kV and 40 mA with the scanning rate of 5° min^−2^. The morphology and microstructure of the product were analyzed by scanning electron microscopy (FESEM, Hitachi, S-4800 II, Tokyo, Japan) and transmission electron microscopy (TEM, Philips, TECNAI 12, Amsterdam, Holland). All X-ray photoelectron spectroscopy (XPS) measurements were carried out on Kratos XSAM-800 spectrometers with an Al Kα radiation source.

### 3.6. Electrochemical Measurements

All of the electrochemical measurements were taken using a three-electrode configuration (CHI 660E electrochemical workstation). The catalyst ink deposited on a glassy carbon electrode (GCE) acted as the working electrode, while a graphite rod and saturated calomel electrode (SCE) worked as the counter and reference electrodes. The preparation of the catalytic ink is as follows: 5 mg of active material, 50 µL of Nafion, and 950 µL of ethanol were mixed under bath sonication for 20 min. Subsequently, 10 µL of the catalyst ink was deposited on the surface of the GCE and dried at room temperature (0.71 mg cm^−2^). The potentials reported in the work were converted to the reversible hydrogen electrode (RHE) by E_(RHE)_ = E_(SCE)_ + 0.0591 * pH + 0.242 V. 

The polarization curves were characterized by cyclic voltammetry (CV) at a scan rate of 5 mV s^−1^, and electrochemical impedance spectroscopy (EIS) was conducted in the frequency range from 100 kHz to 0.01 kHz. Meanwhile, the CV curves in the non-faradic region were measured at scan rates from 5 to 50 mV s^−1^. Chronoamperometry (CA) testing was carried out for 15 h. 

## 4. Conclusions

Herein, we successfully prepared a core–shell-structured CoS_2_@MoS_2_ nanobox through precipitation, sulfurization, and hydrothermal methods. The vertical growth of MoS_2_ nanosheets on the hollow CoS_2_ matrix can effectively inhibit their aggregation and guarantee the exposure of active edge sites. The hollow morphology with a core–shell structure consisting of CoS_2_ and MoS_2_ can accelerate the ion diffusion and charge transfer, while the formed CoS_2_/MoS_2_ heterostructure can boost the number of catalytic active sites during the OER process. Due to the morphological structure and adjusted chemical surface, CoS_2_@MoS_2_ exhibits exceptional catalytic OER performance. To reach the current density of 10 mA cm^−2^, CoS_2_@MoS_2_ only requires the overpotential of 254 mV with a low Tafel slope, small charge transfer resistance, and good catalytic stability.

## Figures and Tables

**Figure 1 molecules-29-01695-f001:**
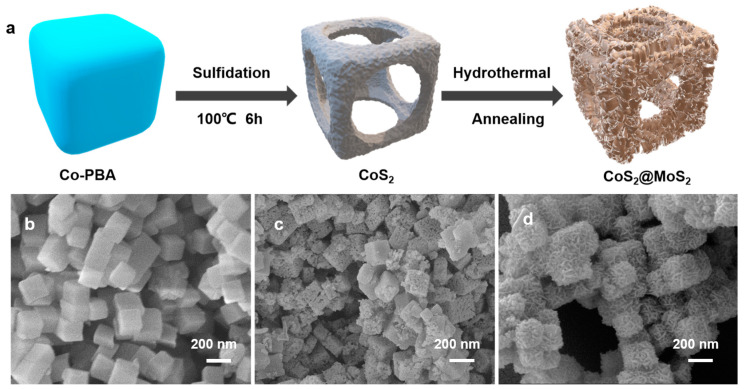
(**a**) Schematic illustration of the synthesis process of CoS_2_@MoS_2_. SEM images of (**b**) Co-PBA, (**c**) CoS_2_, and (**d**) CoS_2_@MoS_2_.

**Figure 2 molecules-29-01695-f002:**
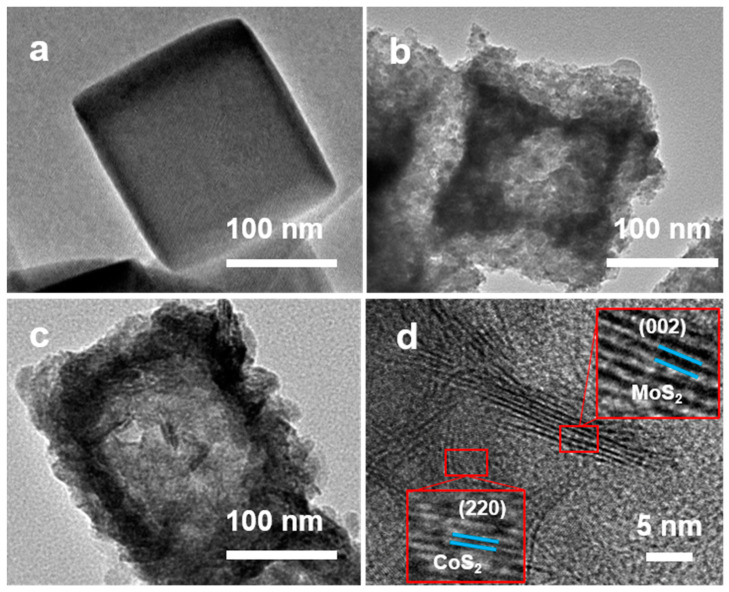
TEM images of (**a**) Co-PBA, (**b**) CoS_2_, and (**c**) CoS_2_@MoS_2_. (**d**) HR-TEM image of CoS_2_@MoS_2_.

**Figure 3 molecules-29-01695-f003:**
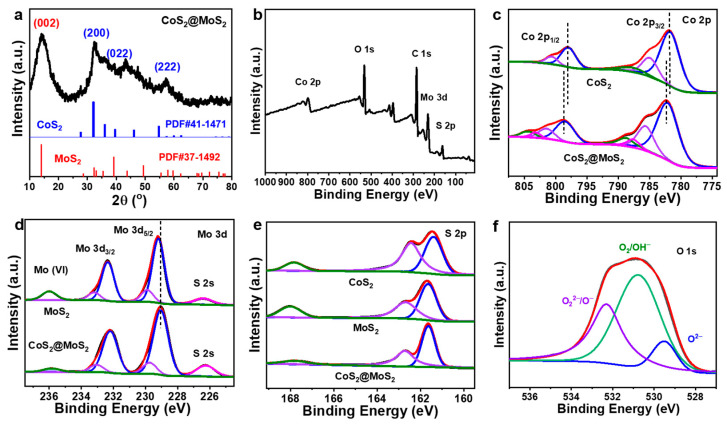
(**a**) XRD pattern of CoS_2_@MoS_2_. (**b**) Full XPS scan of CoS_2_@MoS_2_. High-resolution (**c**) Co 2p of CoS_2_ and CoS_2_@MoS_2_, (**d**) Mo 3d of MoS_2_ and CoS_2_@MoS_2_, (**e**) S 2p of CoS_2_, MoS_2_, and CoS_2_@MoS_2_, and (**f**) O 1s spectra of CoS_2_@MoS_2_.

**Figure 4 molecules-29-01695-f004:**
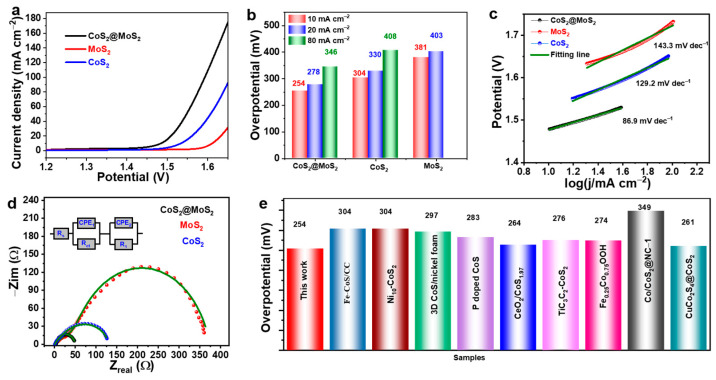
(**a**) Polarization curves of CoS_2_, MoS_2_, and CoS_2_@MoS_2_. (**b**) Overpotential of CoS_2_, MoS_2_, and CoS_2_@MoS_2_ at different current densities. (**c**) Tafel slopes and (**d**) Nyquist plots (inset: the equivalent circuit) of CoS_2_, MoS_2_, and CoS_2_@MoS_2_. (**e**) Comparison of overpotential with the reported results.

**Figure 5 molecules-29-01695-f005:**
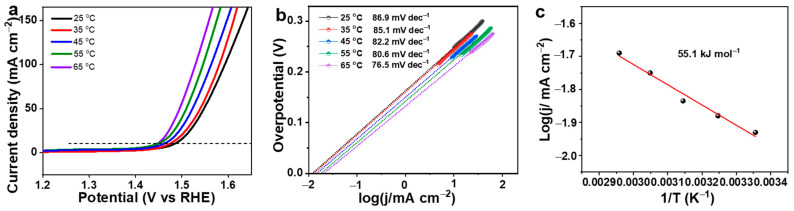
(**a**) Polarization curves and (**b**) Tafel slopes of CoS_2_@MoS_2_ catalyst at different temperatures. (**c**) Arrhenius plots of CoS_2_@MoS_2_ catalyst by linearly fitting 1/T vs. log (j).

**Figure 6 molecules-29-01695-f006:**
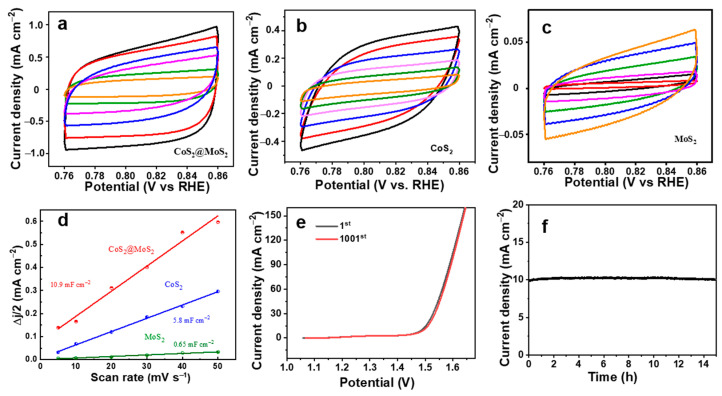
CV curves of (**a**) CoS_2_@MoS_2_, (**b**) CoS_2_, and (**c**)MoS_2_. (**d**) C_dl_ values of CoS_2_, MoS_2_, and CoS_2_@MoS_2_. (**e**) CV curves for CoS_2_@MoS_2_ at 1st and 1001st cycles and (**f**) CA test for 15 h.

**Figure 7 molecules-29-01695-f007:**
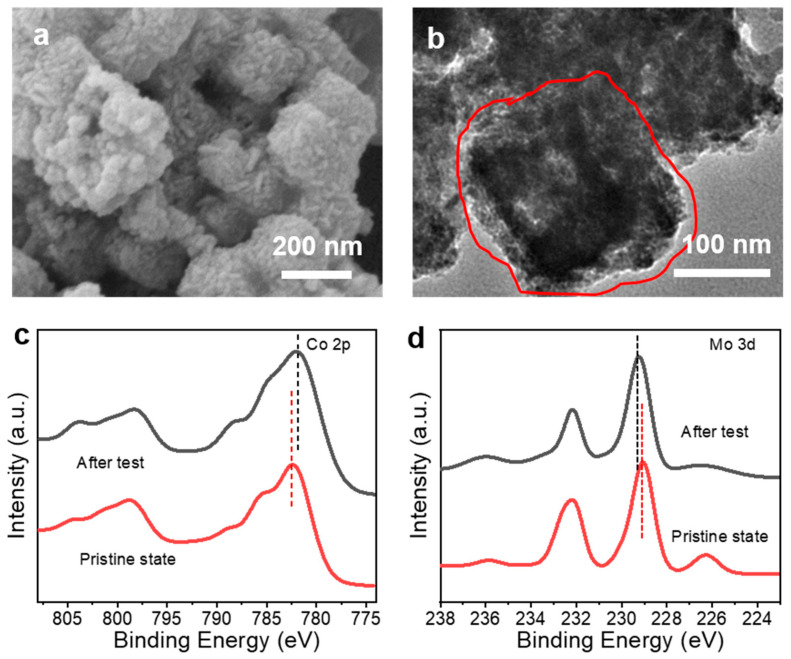
(**a**) SEM and (**b**) TEM images of CoS_2_@MoS_2_ after the CA test. High-resolution XPS spectra of CoS_2_@MoS_2_ before and after the CA test: (**c**) Co 2p and (**d**) Mo 3d.

## Data Availability

Data are contained within the article.

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
