# Peer review of "Core–Shell CoS2@MoS2 with Hollow Heterostructure as an Efficient Electrocatalyst for Boosting Oxygen Evolution Reaction"

_molecules, 2024, doi:10.3390/molecules29081695_

Round 1

Reviewer 1 Report

Comments and Suggestions for Authors

In the submitted manuscript, a core-shell structured CoS2@MoS2 nanocube was synthesized via co-precipitation, followed by low-temperature sulfurization and hydrothermal treatments. The resulting catalyst demonstrates superior OER performance in an alkaline electrolyte. The manuscript meets the rigorous criteria of the journal Molecules, and I recommend its publication after the resolution of several points:

1. The introduction should more strongly highlight the significance of the core-shell structure, referencing pertinent literature to underscore its importance.

2. There are typographical errors present; for instance, 'nanocubic' should be corrected to 'nanocube'.

3. The long-term catalytic stability of the catalyst is a crucial aspect and should be documented with appropriate stability data.

4. A comparative analysis with other reported studies is essential to illustrate the exceptional catalytic OER efficiency of the CoS2@MoS2 nanocube.

Comments on the Quality of English Language

Can be improved.

Reviewer 2 Report

Comments and Suggestions for Authors

This article reported core-shell CoS2@MoS2 with hollow heterostructure for boosting oxygen evolution reaction. Due to the morphological 16 merits and the formation of CoS2/MoS2 heterostructure, CoS2@MoS2 exhibits excellent electrocatalytic performance during the oxygen evolution reaction (OER) process in an alkaline electrolyte. The manuscript can be considered to accept after addressing some remaining issues.

1.      To arouse a broad interest from readership in this field, some strongly related works on sulfide heterostructure for the OER may be useful, such as 10.3390/molecules29020352, 10.26599/NRE.2023.9120106, 10.1007/s12274-023-5670-6, and 10.3390/molecules28207114.

2.      The energy barriers in 1M KOH at different temperatures should be measured and calculated according to the Arrhenius equation.

3.      To explore the structural change, XRD, SEM, and TEM of the catalyst after catalytic OER activity tests should be given to further characterize the used catalyst in detail.

4.      For XPS, it is customary to place the high binding energy on the left side.

5.      The English should be polished by a native English speaker, since there are many grammar and language errors. The title is suggested to change to “Core-shell CoS2@MoS2 with hollow heterostructure as an efficient electrocatalyst for boosting oxygen evolution reaction”.

6.      In the Reference Section, the authors should pay attention to the uniform format, subscripts, case, abbreviation, pages, and full name of the journals. Many small irregularities or inconsistent format appear in the References Sections.

Comments on the Quality of English Language

The English should be polished by a native English speaker, since there are many grammar and language errors.

Reviewer 3 Report

Comments and Suggestions for Authors

In this manuscript, the Donglei Guo and co-authors reported CoS2@MoS2 with hollow heterostructure, exhibiting excellent electrocatalytic performance of oxygen evolution reaction (OER) process in an alkaline electrolyte. To reach the current density of 10 mA cm-2, only 254 mV of overpotential is required for CoS2@MoS2, which is smaller than that of pristine CoS2 and MoS2. Meanwhile, the small Tafel slope (86.9 mV dec-1) and low charge transfer resistance (47 Ω) imply the fast dynamic mechanism of CoS2@MoS2. CoS2@MoS2 shows exceptional catalytic stability. The new results are significant for the rational design of more efficient OER catalysts. Overall, I recommend its publication in Molecules with major revisions. The following comments should be taken for consideration.

1) In this study, authors selected the CoS2@MoS2 electrode. The structure of materials should been analyzed so that to understand the change of CoS2 and MoS2. Please give the related standard card or patterns.

2) CV curves should be compared at the same potential window range in Figure 4a.

3) The color of the corresponding curves of different materials in Figure 4 is inconsistent, which is not conducive to reading.

4) In this paper, the CoS2@MoS2 electrode shows good OER performance. But the relationship between CoS2@MoS2 composite materials and OER activity is no analyzed.

5) The ordinate value is Δcurrent density corresponding to the current difference in Figure 5d, and the corresponding electrode is not marked in curves.

6) Authors propose that CoS2@MoS2 electrode exhibited good stability for OER. Can the author prove it by means of characterization? For example, XRD for crystal structure analysis or SEM for morphology analysis?

7) In this paper, XPS of Co element after CA test showed the shift toward low binding energy, corresponding to the intermediates of cobalt oxides or oxyhydroxides. It is well known that the CV curve of cobalt oxides or oxyhydroxides should have oxidation peaks during OER.

8) Please continue to modify the language expression and write error. For example, “Synthesis of hollow CoS2@ReS2 in P 8; “Current Density or Current density in all figures…

Comments on the Quality of English Language

Minor editing of English language required

Reviewer 4 Report

Comments and Suggestions for Authors

In this work, the authors prepare core-shell CoS2@MoS2 composite with hollow heterostructure for boosting OER, which achieve a good activity and stability. The quality of this manuscript is good and can be accepted after well addressing the comments below.

1.     The oxygen species in the materials are important for the OER activity, such as adsorbed oxygen, oxygen vacancies, and lattice oxygen. The authors should fit and analyze the O 1s XPS to demonstrate the performance origin. Please refer to 10.1002/cey2.465 for this point.

2.     Some necessary performance comparisons with other reported sulfides should be added to demonstrate the good performance of their catalysts. The authors can refer to this work 10.1016/j.cej.2023.141674 for comparisons.

3.     The morphology and structure of the catalyst after OER should be analyzed by SEM and XRD, respectively.

4.     The synergetic effects between CoS2 and MoS2 should be well discussed to further explain the underlying mechanism.

Comments on the Quality of English Language

Minor editing of English language required

Round 2

Reviewer 2 Report

Comments and Suggestions for Authors

The revised manuscript can be accepted now.